# The First Complete Mitogenome Characterization and Phylogenetic Implications of *Elops machnata* (Teleostei: Elopiformes: Elopidae)

**DOI:** 10.3390/biology14070739

**Published:** 2025-06-21

**Authors:** Jia-Yu Li, Xue-Fan Cui, Shu-He Chen, Yang Li, Shui-Yuan Zhang, Yuan-Feng Yang, Yi-Yang Li, Yu-Song Guo, Zhong-Duo Wang, Jian Liao

**Affiliations:** 1Key Laboratory of Aquaculture in South China Sea for Aquatic Economic Animals of Guangdong Higher Education Institutes, College of Fisheries, Guangdong Ocean University, Zhanjiang 524088, China; lijiayu@stu.gdou.edu.cn (J.-Y.L.); liyang@stu.gdou.edu.cn (Y.L.); zsy231@stu.gdou.edu.cn (S.-Y.Z.); gdouyyf228@stu.gdou.edu.cn (Y.-F.Y.); lyy108@stu.gdou.edu.cn (Y.-Y.L.); gysrabbit@163.com (Y.-S.G.); wangzd@gdou.edu.cn (Z.-D.W.); 2Guangdong Provincial Key Laboratory of Aquatic Animal Disease Control and Healthy Aquaculture, College of Fisheries, Guangdong Ocean University, Zhanjiang 524088, China; 3State Key Laboratory of Mariculture Breeding, Key Laboratory of Marine Biotechnology of Fujian Province, College of Marine Sciences, Fujian Agriculture and Forestry University, Fuzhou 350002, China; cuixuefan@fafu.edu.cn; 4Graduate School of Asian and African Area Studies, Kyoto University, Kyoto 6068501, Japan; chin.jiuka.3i@kyoto-u.ac.jp

**Keywords:** *Elops machnata*, complete mitogenome, protein-coding genes, Elopomorpha, phylogenetic

## Abstract

This study reports the first complete mitochondrial genome of *Elops machnata*, a primitive teleost fish important for studying early ray-finned fish evolution. The 16,712 bp mitogenome contains the standard 37 genes and exhibits a slight AT bias (52.53%). Codon usage analysis revealed the preferred codons, while the tRNA structures mostly followed typical patterns, with the exception of *trnS1*(gct). Phylogenetic analysis confirmed the monophyly of the four major Elopomorpha groups (Notacanthiformes, Albuliformes, Anguilliformes, and Elopiformes). Additionally, phylogenetic analyses validated a close relationship between *E. machnata* and *E. hawaiensis*. These findings provide valuable genomic data for understanding teleost evolution and mitochondrial diversity in this ancient fish group.

## 1. Introduction

Elopomorpha, a highly diversified clade within Teleostei fish, encompasses multiple lineages, including Anguilliformes (eels), Saccopharyngiformes (gulper eels), Notacanthiformes (halosaurs and spiny eels), Elopiformes (tarpons and ladyfishes), and Albuliformes (bonefishes) [1]. Among these, Elopiformes represents one of the most ancient teleost orders, comprising the extant families Elopidae and Megalopidae alongside several extinct taxa, illustrating a rich evolutionary history [2]. Elopidae, the most species-rich family within this order (containing seven extant species), is characterized by the genus *Elops*, which exhibits a distinctive leptocephalus larval stage, a rare developmental feature of significant value in fish evolutionary studies [3,4,5,6].

*Elops machnata*, a member of the genus *Elops*, is widely distributed in Indo-Pacific coastal waters, being particularly abundant in Southeast Asia and East African seas. As a keystone species in estuarine ecosystems, it plays a vital role in commercial fisheries, recreational angling, and local livelihoods [3,6]. Given its ecological significance, understanding the genetic basis underlying its biology becomes crucial. Mitochondrial DNA, with its unique inheritance pattern and high evolutionary conservation in certain regions, serves as an ideal molecular marker for such investigations. While preliminary analyses of mitochondrial gene fragments and regional genetic diversity have been performed [7,8], the species’ evolutionary mechanisms, population genetic structure, and conservation strategies remain poorly understood due to limitations in genomic data completeness. Concurrently, phylogenetic relationships within Elopomorpha have long been debated. Early studies questioned its monophyly [9,10], whereas contemporary multigene analyses consistently support its monophyletic origin [11,12,13,14]. This debate holds significant importance because resolving the phylogeny of Elopomorpha can provide crucial insights into broader fish evolutionary questions. Elopomorpha represents an ancient and diverse lineage of fish, and understanding its evolutionary history helps reconstruct the overall evolutionary tree of fish. By clarifying its phylogenetic relationships, researchers can better trace the evolution of key morphological, physiological, and ecological traits across different fish groups, which in turn contributes to a more comprehensive understanding of vertebrate evolution. However, the timing of Elopiformes’s origin, evolutionary pathways, and taxonomic frameworks require further elucidation [2,15]. Against this backdrop, systematic genomic investigations of *E. machnata* using complete mitogenome sequencing not only promise insights into its population genetics and adaptive evolution but also hold critical scientific value for resolving phylogenetic controversies within Elopomorpha.

Mitochondria, eukaryotic cellular powerhouses originating from endosymbiotic events [16], generate ATP via oxidative phosphorylation to sustain cellular metabolism [17]. Their maternal inheritance, small genome size, and rapid evolutionary rates render them ideal markers for evolutionary biology. Recent studies revealed paternal mitochondrial inheritance in specific lineages [18], expanding research dimensions for genetic mechanisms. Typical teleost mitochondrial genomes are circular double-stranded DNA molecules, encoding 13 protein-coding genes, 22 tRNA genes, 2 rRNA genes, and a non-coding control region (*D-loop*) regulating genome replication and expression [19,20,21]. Varied evolutionary rates across regions enable multiscale phylogenetic analyses [22]. Advances in sequencing technologies and bioinformatics have popularized mitochondrial genome data in species identification, phylogeography, and population genetics due to their accessibility and cost-effectiveness [23,24,25,26].

Notably, *E. machnata* is classified as “data deficient” (DD) by the IUCN [3], highlighting critical gaps in understanding its genetic diversity, population structure, and evolutionary history. This data scarcity directly impedes evidence-based conservation strategies and phylogenetic resolution within Elopomorpha. In particular, systematic characterization of its complete mitochondrial genome structure remains lacking, significantly hindering the understanding of population dynamics, adaptive evolution, and phylogenetic relationships. By addressing this knowledge gap through the first complete mitochondrial genome sequencing, assembly, and annotation of *E. machnata* using high-throughput technologies, this study directly contributes to filling the IUCN-identified data deficiency. Comparative analyses with 23 Elopomorpha mitochondrial genomes from the NCBI, employing maximum likelihood and Bayesian methods, reconstruct phylogenetic trees and estimate divergence times to elucidate evolutionary statuses. These findings not only provide crucial data supporting taxonomic revisions, conservation biology assessments, and phylogenetic studies of Elopomorpha species but also establish a genomic foundation for future population genetics and conservation efforts on this data-deficient species.

## 2. Materials and Methods

### 2.1. Sample Collection and DNA Extraction

Samples were collected from Huguang Town, Mazhang District, Zhanjiang City, Guangdong Province, China (110.302623° E, 21.092811° N). Tail muscle tissues were excised and immediately immersed in 95% ethanol-containing centrifuge tubes to inhibit microbial growth and prevent DNA degradation, followed by storage at −20 °C until analysis. Genomic DNA was extracted using a commercial kit (Sangon Biotech, Shanghai, China) according to the manufacturer’s standardized protocol. To ensure the quality of the sequencing data, the extracted DNA samples underwent rigorous quality control (QC) evaluations, including concentration measurement using a NanoDrop spectrophotometer (Thermo Fisher Scientific, Waltham, MA, USA), purity analysis via absorbance ratio of 260 nm/280 nm (A260/A280, ideal range of 1.8–2.0 for assessing protein contamination) and 260 nm/230 nm (A260/A230, acceptable range of 1.8–2.2 for evaluating salts, organic solvents, or polysaccharide contamination), and integrity verification through agarose gel electrophoresis (1.0–1.5% gel with 0.5 μg/mL SYBR Green dye, expecting a single major band at ~16 kb). Only samples meeting these high-throughput sequencing requirements were processed for downstream analyses. By systematically integrating morphological classification, DNA barcoding, and refined DNA extraction techniques, this study established a full-chain QC system encompassing specimen collection, genetic material extraction, and molecular identification, effectively safeguarding the biological representativeness of the research materials and accuracy of the genetic information.

### 2.2. Sequencing and Assembly

Sequencing libraries were constructed using an Illumina TruSeq™ Nano DNA Sample Preparation Kit (Illumina, San Diego, CA, USA) to process the extracted total genomic DNA, with an insert size of 450 bp to optimize sequencing coverage. The libraries were sequenced on the Illumina HiSeq platform (Illumina, San Diego, CA, USA), generating 150 bp paired-end reads to ensure a minimum of 6 GB of raw data per library, providing a sufficient sequencing depth for genome assembly. To rigorously control data quality, the Trimmomatic tool was employed for multi-round filtering: (1) removing adapter sequences and PCR duplicates by (2) trimming bases with quality scores below Q20; and (3) filtering reads containing 10% or more unidentified bases (Ns). Regarding nuclear mitochondrial DNA segments (NUMTs), several steps were taken to remove them from the sequencing data. First, we used BLASTn to search the raw reads against a database of known NUMTs from related species. Reads with high-scoring alignments (e value < 1 × 10⁻¹⁰ to the NUMT database were flagged. Then, we employed a de novo assembly-based approach. After the initial assembly with SPAdes version 3.15.5, we compared the assembled contigs to the reference mitochondrial genome of closely related species. Contigs that showed significant divergence in terms of sequence characteristics (e.g., different GC content or abnormal gene arrangements) and had low-identity alignments with the reference mitochondrial genome were considered potential NUMTs and removed. This two-step approach effectively minimized the presence of NUMTs in our final mitochondrial genome dataset. The cleaned data obtained after quality control were used for de novo genome assembly using the SPAdes assembler software version 3.15.5, with the default parameters and algorithm optimization tailored to mitochondrial genome characteristics to generate high-accuracy contiguous contig sequences. This workflow, through standardized library construction, stringent data quality control, and parameter-optimized assembly, ensured the reliability and accuracy of the mitochondrial genome research.

### 2.3. Annotation and Sequence Analysis

In this study, annotation of the assembled *E. machnata* mitochondrial genome was performed using the Galaxy Web Server platform [27], combined with the tRNAscan-SE tool to assist with the identification of ribosomal RNA (rRNA) and transfer RNA (tRNA) genes [28]. Furthermore, the predicted start and stop sites of the rRNA, tRNA, and protein-coding genes were manually calibrated using the verified mitochondrial genome data of Elopidae species in the NCBI database as a reference to improve annotation accuracy. The genome structure was visually displayed as a circular diagram using OrganellarGenomeDRAW (OGDRAW) v1.3.1 software [29]. The Compute Nucleotide Composition module of MEGA software version 7 [30] was used to calculate the base composition and analyze the base distribution bias by computing the AT-skew ((A − T)/(A + T)) and GC-skew ((G − C)/(G + C)). Additionally, the Compute Code Usage Bias function was used to calculate the frequency of amino acid usage and relative synonymous codon usage (RSCU) for 13 protein-coding genes, and the results were visualized using the aplot and ggplot2 packages in R to reveal codon usage preferences. The average rates of nonsynonymous substitutions (dN), synonymous substitutions (dS), and dN/dS ratios for each PCG were calculated using DnaSP v6.12 [31]. Furthermore, the amino acid composition of mitochondrial protein-coding genes in each species was calculated using the Compute Amino Acid Composition module of MEGA software version 7 and visualized using TBtools v2.142 [32] to provide data support for subsequent genomic analysis.

### 2.4. Phylogenetic Tree Reconstruction

To reconstruct the phylogenetic relationships, we obtained the complete mitochondrial genome data of 22 Elopomorpha species from the NCBI database, downloaded the mitochondrial genome of *Sander vitreus* (Perciformes, Percidae) as the outgroup, and combined it with the self-assembled complete mitochondrial sequence of *E. machnata* for phylogenetic analysis (Appendix A). We downloaded all publicly available mitochondrial genome sequences of the superorder Elopomorpha from the NCBI database up to 26 December 2024. For the four orders retrievable within this superorder (Notacanthiformes, Albuliformes, Anguilliformes, and Elopiformes), we selected at least one complete mitochondrial genome sequence from each family, adhering to the principle of phylogenetic representativeness. This sampling strategy ensures that the selected dataset covers the major phylogenetic lineages and ecological diversities within Elopomorpha. Thirteen protein-coding genes (PCGs) were extracted from the mitochondrial genomes for phylogenetic analysis. PhyloSuite software v1.2.3 was used to construct phylogenetic trees using both the maximum likelihood (ML) and Bayesian inference (BI) methods [33]. Prior to phylogenetic tree construction, protein-coding sequences were aligned using MAFFT implemented in PhyloSuite v1.2.3. Alignments were then optimized with MACSE (for CDS) and trimmed using Gblocks to remove poorly aligned regions [33]. For the construction of the maximum likelihood tree, inference was performed under the Edge-linked partition model using the IQ-TREE v2.2.0 program [34] within PhyloSuite software v1.2.3, and 5000 ultrafast bootstrap replicates [35] were performed to assess node support. Prior to Bayesian tree construction, the ModelFinder v2.2.0 tool [36] was used to select the best-fit partition model (Edge-linked) based on the BIC criterion, ultimately determining GTR + F + I + G4 to be the optimal model. Bayesian inference was performed using MrBayes v3.2.7a software [37], with two parallel runs each running for 300 million generations, a sampling interval of 1000 generations, and the first 25% of the samples discarded as burn-in.

## 3. Results

### 3.1. Genome Organization and Composition

The mitochondrial genome of *E. machnata* determined in this study exhibited a typical circular structure (16,712 bp total length), encoding a total of 37 genes (13 protein-coding, 22 tRNA, 2 rRNA, and 1 control region (CR)). Among these, 9 genes, including *trnP*(tgg), *trnE*(ttc), *nad6*, *trnS2*(tga), *trnY*(gta), *trnC*(gca), *trnN*(gtt), *trnA*(tgc), and *trnQ*(ttg), were located on the heavy strand (H-strand), while the remaining 28 genes were concentrated on the light strand (L-strand) (Figure 1 and Table 1). Base composition analysis revealed that the genome-wide A, T, G, and C contents were 22.71%, 29.82%, 30.11%, and 17.36%, respectively, with an overall AT content of 52.53%, indicating a slight AT bias. The CR spanned 968 bp and had an AT content as high as 63.33%, significantly higher than the genome-wide average and consistent with the typical characteristics of vertebrate mitochondrial control regions. In terms of gene arrangement features, the genome demonstrated a compact organizational structure, comprising 15 seamless gene junctions, 10 gene overlaps (with overlap lengths ranging from 1 to 10 bp), and 12 gene intervals. It was noteworthy that there was an interval sequence as long as 39 bp between *rrnL* and *trnV*(tac), while the interval between *trnS1*(gct) and *trnE*(ttc) was only 1 bp, reflecting the evolutionary plasticity of gene arrangement. Further comparative analysis of mitochondrial gene orders among 23 species of Elopomorpha revealed that except for *Ophichthus erabo* and *Pterothrissus gissu*, which exhibited a reciprocal translocation between the *cob*-*trnT* and *nad6*-*trnE* gene clusters, and another between the *trnL1* and *trnH*-*trnS1* gene clusters, the gene orders of the remaining species were highly conserved (Figure 2). This result supports the evolutionary homology and stability of Elopomorpha.

### 3.2. Protein-Coding Genes

In this study, the base skewness exhibited an A-T skew of 0.052 and a C-G skew of 0.291. It is noteworthy that the *nad6* gene demonstrated a unique base distribution pattern, with a T content as high as 40%, significantly deviating from other protein-coding genes (T content = 20–28%), while the A content was only 12%, forming a stark contrast in *E. machnata* (Figure 3A). However, our further analysis of the T base composition in the *nad6* genes of 23 Elopomorpha species revealed that except for the two species in the genus *Albula*, all other species showed similarity to *E. machnata* (i.e., the T base content approached 40%). The analysis of codon usage bias revealed that the mitochondrial genome of this species follows a typical vertebrate codon usage pattern. In terms of start codons, except for the *cox1* gene, which uses GTG as the start codon, the remaining 12 genes all use the standard ATG start codon. The analysis of the termination codon usage frequency showed that TAA was the most frequently used termination codon (accounting for 53.8%). Through RSCU value analysis (Figure 4), it was revealed that CGA (arginine), CUA (leucine), GCC (alanine), and GGA (glycine) were high-frequency codons, with RSCU values reaching 2.44, 2.36, 1.88, and 1.84, respectively. It is noteworthy that the third site of these four dominant codons is the A base, indicating a significant preference for the A base at the third codon site in the mitochondrial genome of *E. machnata*. We calculated the dN/dS ratios, representing the ratio of the nonsynonymous substitution rate (dN) to the synonymous substitution rate (dS), for 13 protein-coding genes (PCGs) in the mitochondrial genomes of 23 Elopomorpha species (Figure 5). The results showed that the dN/dS ratios of all 13 PCGs were less than one, indicating purifying selection. This suggests that the functions of these PCGs are crucial, and variations in their amino acid sequences are strongly constrained by natural selection. The evolutionary rates (dN/dS ratios) of the 13 PCGs are as follows: *nad1* (0.5108) > *nad6* (0.2242) > *cox1* (0.2230) > *nad5* (0.2214) > *nad4* (0.1460) > *nad2* (0.1432) > *atp8* (0.1306) > *nad4L* (0.0928) > *cytb* (0.0684) > *nad3* (0.0573) > *atp6* (0.0451) > *cox3* (0.0396) > *cox2* (0.0275). The heatmap analysis of the amino acid composition (Figure 6) further revealed that Elopomorpha species generally exhibited high-frequency usage of leucine (Leu) in protein-coding genes (RSCU 2.44), followed by alanine (Ala) (RSCU 1.88), while the usage frequencies of other amino acids were relatively low. This characteristic of amino acid composition may be closely related to the functional adaptability and evolutionary pressure of mitochondrial proteins, reflecting the conservation and specificity of Elopomorpha in the process of molecular evolution.

### 3.3. Transfer and Ribosomal RNA Gene

We analyzed the 22 transfer RNA (tRNA) genes of the mitochondrial genome of *E. machnata*. Their sequence lengths ranged from 68 to 75 bp, with *trnL2*(taa) being the longest (75 bp) and *trnC*(gca) being the shortest (68 bp). The total length of the 22 tRNA*s* was 1562 bp. The combined AT content was 55.56%, showing a significant AT bias. In terms of spatial distribution, eight tRNA genes, including *trnP*(tgg), *trnE*(ttc), *trnS2*(tga), *trnY*(gta), *trnC*(gca), *trnN*(gtt), *trnA*(tgc), and *trnQ*(ttg), were located on the H-strand, while the remaining 14 tRNA genes were distributed on the L-strand. Through secondary structure prediction, 21 tRNAs could fold into a typical cloverleaf structure, but *trnS1*(gct) had a structural variation, lacking the dihydrouridine arm (DHU arm) (Appendix A). Further analysis of the *trnS1*(gct) secondary structures across 22 Elopomorpha species revealed that 9 of them exhibited a similar DHU arm deletion. These species were *A. glossodonta*, *A. vulpes*, *E. hawaiensis*, *E. saurus*, *G. odishi*, *M. atlanticus*, *M. cyprinoides*, *P. rissoanus*, and *P. gissu* (Figure 7). It is noteworthy that 13 tRNAs exhibited extensive wobble base pairings. These non-canonical base pairings were mainly distributed in the acceptor arm (eight cases), TΨC arm (seven cases), anticodon arm (seven cases), and D arm (six cases), and they may participate in the regulation of mitochondrial translation by enhancing structural flexibility. In terms of ribosomal RNA (rRNA), this genome contained two components—*12S rRNA* and *16S rRNA*—both located on the light strand, with lengths of 957 bp and 1663 bp, respectively. They were separated by the *trnV*(tac) gene. As core components of the mitochondrial protein synthesis system, these two rRNA molecules collaborate with ribosomal proteins through specific spatial arrangements to construct the catalytic center of mitochondrial ribosomes, ensuring the efficiency and accuracy of mitochondrial gene expression.

### 3.4. Phylogenetic Analysis of Elopomorpha Species

We constructed high-resolution phylogenetic trees using both the maximum likelihood (ML) and Bayesian inference (BI) methods, based on the 13 protein-coding genes of the mitochondrial genomes of 23 Elopomorpha species and the outgroup *Sander vitreus* (OL477730). The molecular trees generated by different methods were highly consistent in terms of overall topology, with discrepancies only in the placement of *Halosaurus ovenii* within the Notacanthiformes clade, reflecting the complex evolutionary history of this order (Figure 8). Combined phylogenetic analysis revealed the monophyly of the four major Elopomorpha groups (Notacanthiformes, Albuliformes, Anguilliformes, and Elopiformes) and their sister group relationships, specifically (((Notacanthiformes + Albuliformes) + Anguilliformes) + Elopiformes) (Figure 7).

## 4. Discussion

### 4.1. E. machnata Mitochondrial Genome Characteristics and Their Implications

In this study, we analyzed the complete mitochondrial genome (16,712 bp) of *E. machnata* using bioinformatics tools. The AT content was found to be 52.53%, which aligns with the typical AT bias characteristic of fish mitochondrial genomes, as observed in species such as the Fasin rainbow fish (*Melanotaenia fasinensis*) [24], Parkinson’s Rainbowfish (*Melanotaenia parkinsoni*) [38], and *Odontobutis* spp [39]. This proportion is highly similar to those of closely related species within the Elopidae family, such as *E. saurus* and *E. hawaiensis*, indicating a significant conservation in base composition among Elopidae species. The AT bias, a common phenomenon in fish mitochondrial genomes, may be closely related to mitochondrial DNA replication mechanisms, repair efficiency, and energy metabolism requirements [40]. Its stability across different species reflects the selective pressures acting on mitochondrial genomes during evolution. Mitochondrial DNA sequences and their genomic structures serve as crucial molecular markers, demonstrating unique advantages in evolutionary studies [41]. Our analysis revealed that among the 23 Elopomorpha species examined, only *O. erabo* exhibited gene rearrangement, further confirming the high conservation of the mitochondrial genome gene order in Elopomorpha [42]. This conservation may stem from the physical linkage requirements of mitochondrial functional genes and the pressure of functional co-evolution. The gene rearrangement event in *O. erabo* may provide clues for exploring environmental adaptation or special evolutionary pathways. For instance, in some fish species in Antarctica, there are large blocks of gene inversion in their mitochondrial genomes, involving multiple genes and control regions [43]. This inversion might be related to their adaptation to the special environment with a low temperature and high salinity in Antarctica. An analysis of tRNA secondary structures revealed that 21 tRNAs exhibited the typical cloverleaf configuration, while *trnS1*(gct) lacked the dihydrouridine arm (DHU arm), a feature widely observed in fish and potentially associated with tRNA functional simplification or special translation mechanisms [44,45]. Codon usage bias showed that CUA (L), CGA (R), GCC (A), and GGA (G) were high-frequency codons, all with RSCU values greater than one, reflecting a significant preference for A bases at the third codon position in the mitochondrial genome of *E. machnata*. This preference may be related to the optimization of mitochondrial translation system efficiency and the minimization of energy metabolism costs [46]. Studies have shown that preferred codons exhibit a higher translation speed and accuracy in *Drosophila melanogaster* [47].

### 4.2. Elopomorpha Phylogeny: Evolutionary Reconstruction and Relationships

This study revealed the conserved features of mitochondrial genomes in 23 Elopomorpha species. The gene arrangements of the vast majority of species were highly consistent, providing molecular evidence for the monophyly in the evolutionary history of Elopomorpha. Notably, genome rearrangements existed in individual species such as *O. erabo* and *P. gissu*, which might be related to their higher evolutionary rates and resulted from the combined drive of natural selection and adaptive evolution [48]. Further analysis showed that the amino acid compositions of the 13 protein-coding genes (PCGs) in mitochondrial genomes of different species were significantly consistent, suggesting that these genes played core roles in energy metabolic pathways and their sequence variations were strictly constrained by natural selection [23]. In terms of tRNA structure, the *trnS1* gene of most species lacked the DHU arm, forming an adaptively simplified cloverleaf structure. This feature might be related to the “simplification” evolutionary strategy of mitochondrial tRNAs, compressing the genome length by deleting non-essential domains (teleost mitochondrial genomes generally had compact gene arrangements) [48]. It is noteworthy that subtle differences in this structure existed among different orders and families, possibly reflecting lineage-specific evolutionary trajectories [49]. In phylogenetic analysis, the genus *Albula* exhibited a significant long-branch phenomenon, indicating that this group might have experienced a rapid evolutionary process. This phenomenon was corroborated in two aspects. Firstly, rapid differentiation at the genomic level might have been related to driving factors such as environmental changes and niche differentiation [48]. Secondly, base composition analysis of the *nad6* gene showed that the base T content in the vast majority of the 23 species was approximately 40%, while that in *A. glossodonta* and *A. vulpes* of the genus *Albula* was only about 30%, further supporting the specific evolutionary pattern of this genus. The conserved mitochondrial gene architecture and specific evolutionary signatures (such as AT bias) observed in *E. machnata* provide a robust molecular foundation for reconstructing phylogenetic relationships within Elopomorpha. The phylogenetic tree of Elopomorpha constructed in this study based on mitochondrial genome data clarified the phylogenetic position of *E. machnata* for the first time. The results show that *E. machnata* and *E. hawaiensis* formed a close clade, with their genetic distance supporting the hypothesis of their recent divergence [6]. Further analysis of *COI* gene sequences (650 bp fragment) from public databases revealed intraspecific genetic distances between the two species as low as 0.25%, and haplotype network analysis indicated shared haplotypes between them [6]. Additionally, the monophyletic clade formed by *E. machnata* + *E. hawaiensis* in the ultrametric tree showed a posterior probability of >0.9, consistent with the low genetic divergence and geographic sympatry observed in the Indo-Pacific region [6]. These multi-methodological results (phylogenetic tree, genetic distance, and haplotype sharing) collectively support the hypothesis of recent speciation or potential synonymy between the two taxa. The multi-gene concatenation analysis used in this study significantly improved the resolution of the phylogenetic tree, overcoming the limitations of single-gene studies (such as *Cytb*, *COI*) [7]. The mitochondrial genome data provided by this study lays the foundation for research on the differentiation of Elopiformes species. In the future, it will be necessary to combine nuclear genome data to reveal a more complete evolutionary history. Furthermore, the improvement of mitochondrial genome databases, such as the addition of *E. affinis* data, will contribute to a comprehensive understanding of the evolutionary mechanisms of Elopomorpha.

Additionally, in the phylogenetic tree of this study, the species *H. ovenii* is classified under the family Halosauridae within the order Notacanthiformes according to traditional taxonomy. The order Notacanthiformes comprises two families: Halosauridae and Notacanthidae [50,51]. Previous studies have indicated that *H. ovenii* shares a common ancestor with species in the family Notacanthidae and is closely related to species in the family Halosauridae [52]. Bañón et al. (2016) conducted a comprehensive taxonomic study on species within the family Halosauridae, revealing consistency between morphological and molecular data [50]. Barros-García et al. (2018) integrated morphological and molecular data to perform a phylogenetic analysis and time calibration of the order Notacanthiformes, but neither of these studies mentioned the evolutionary relationship of *H. ovenii* [53]. In the phylogenetic tree constructed using the Bayesian method in this study, *H. ovenii* exhibited a closer relationship with species in the family Halosauridae, whereas in the tree constructed using the maximum likelihood method, it showed a closer relationship with species in the family Notacanthidae. Given that the current taxonomy classifies *H. ovenii* as a species within the family Halosauridae, the aforementioned discrepancies raise the question of whether the classification of *H. ovenii* at the family level requires revision. Therefore, we propose that *H. ovenii* may represent an intermediate species existing between the families Halosauridae and Notacanthidae. Similar evolutionary controversies exist within the suborder Anisozygoptera of the order Odonata, for which no definitive classification criteria have been established [23,54]. Given the limitations of mitochondrial whole genomes in phylogenetic and taxonomic analyses, it remains inconclusive whether *H. ovenii* is an intermediate species between the families Halosauridae and Notacanthidae. We recommend that the scientific community undertake more rigorous and meticulous work on the classification of *H. ovenii*, including but not limited to analyses at the whole-genome level, incorporating more variant information and integrating additional fossil data to elucidate its evolutionary trajectory. Mitochondrial DNA (MtDNA) represents a single, maternally inherited locus that may not reflect the true evolutionary history of a species due to processes like introgression, selective sweeps, or incomplete lineage sorting. MtDNA can be transferred between species via hybridization, leading to discordance between mitochondrial and nuclear phylogenies. MtDNA may not provide sufficient variation to resolve rapid radiations or recent speciation events [23,49]. Nuclear markers (e.g., RADseq and ultraconserved elements) would complement mtDNA as follows. (1) They would provide multiple independent loci. Nuclear genomes offer thousands of unlinked markers, reducing the effects of stochastic lineage sorting and improving phylogenetic accuracy [23]. (2) Nuclear data can identify admixed genomic regions using methods like ABBA-BABA tests, which is critical for distinguishing between shared ancestry and gene flow [48]. (3) Finally high-resolution nuclear datasets are better suited to inferring relationships within recently diverged clades [49]. Therefore, integrating mitochondrial and nuclear data represents a superior approach for analyzing species evolutionary questions.

## 5. Conclusions

This study represents the first complete mitochondrial genome of *E. machnata*, deposited in GenBank under accession number PV294982. The circular genome spans 16,712 bp and exhibits pronounced AT bias (52.53% overall, with the control region reaching 63.33%). Comparative genomic analysis across 23 Elopomorpha species revealed a highly conserved gene order, with only *O. erabo* showing a gene translocation, underscoring the homologous stability of this taxonomic group during evolution. The base composition of protein-coding genes conforms to vertebrate patterns (A-T skew of 0.052 and C-G skew of 0.291), yet the *nad6* gene uniquely exhibited an elevated T-base frequency (40%). RSCU analysis identified CGA, CUA, GCC, and GGA as high-frequency codons, with significant A-bias at the third codon position. The combined high usage of leucine (Leu) and alanine (Ala) suggests potential functional optimization and evolutionary selection pressures. tRNA secondary structure analysis showed 21 tRNAs possessing canonical cloverleaf configurations, except for *trnS1*(gct) lacking the DHU arm. Extensive wobble base pairings were observed in 13 tRNAs, with non-canonical pairing patterns distributed across the acceptor, TΨC, anticodon, and D arms, potentially enhancing structural flexibility to modulate translational efficiency. Phylogenetic analysis revealed the monophyly of the four major Elopomorpha groups (Notacanthiformes, Albuliformes, Anguilliformes, and Elopiformes) and their sister group relationships, specifically (((Notacanthiformes + Albuliformes) + Anguilli-formes) + Elopiformes). This study established the first high-resolution temporal framework for Elopomorpha evolution, providing critical genetic insights into their adaptive radiation mechanisms.

## Figures and Tables

**Figure 1 biology-14-00739-f001:**
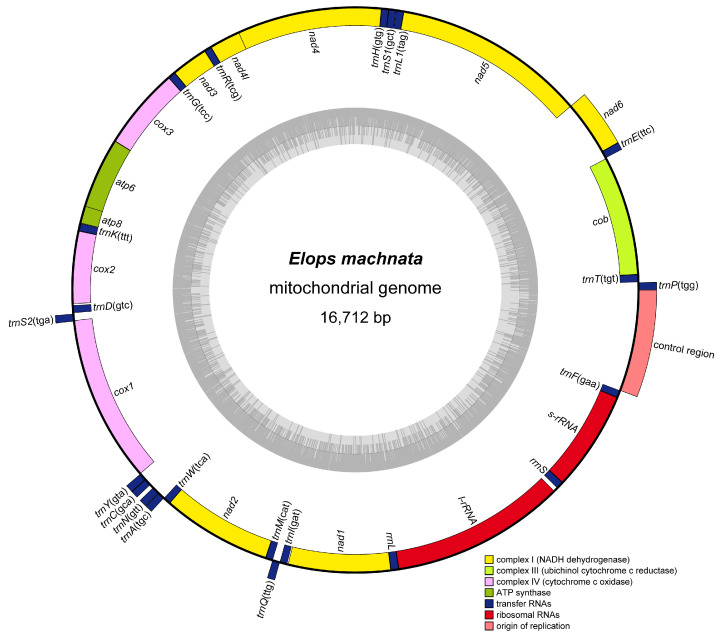
Circular structure diagram of the mitochondrial genome of *E. machnata*, with the outer circle representing the heavy strand (H), the inner circle representing the light strand (L), the outer gray circle denoting the distribution of AT, and the inner gray circle denoting the distribution of the GC content.

**Figure 2 biology-14-00739-f002:**
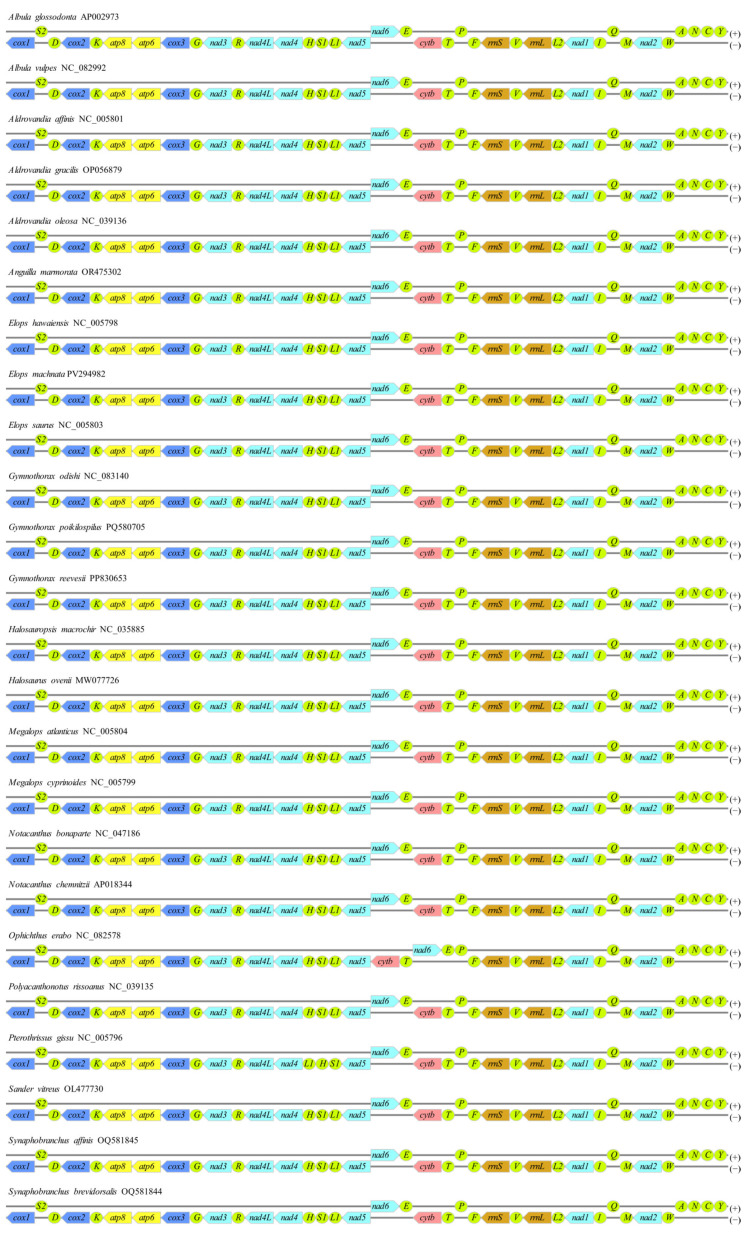
Mitochondrial genome gene orders of 23 species of Elopomorpha and the outgroup *Sander vitreus*. Here, *cox1*-*cox3*, *nad1*-*nad6* and *nad4L*, *atp6* and *atp8*, *cytb*, tRNA*s*, and rRNAs are displayed in dark blue, light blue, yellow, pink, green, and orange, respectively. Except for tRNAs, which are oval-shaped, the remaining genes are shown as left-pointing arrows.

**Figure 3 biology-14-00739-f003:**
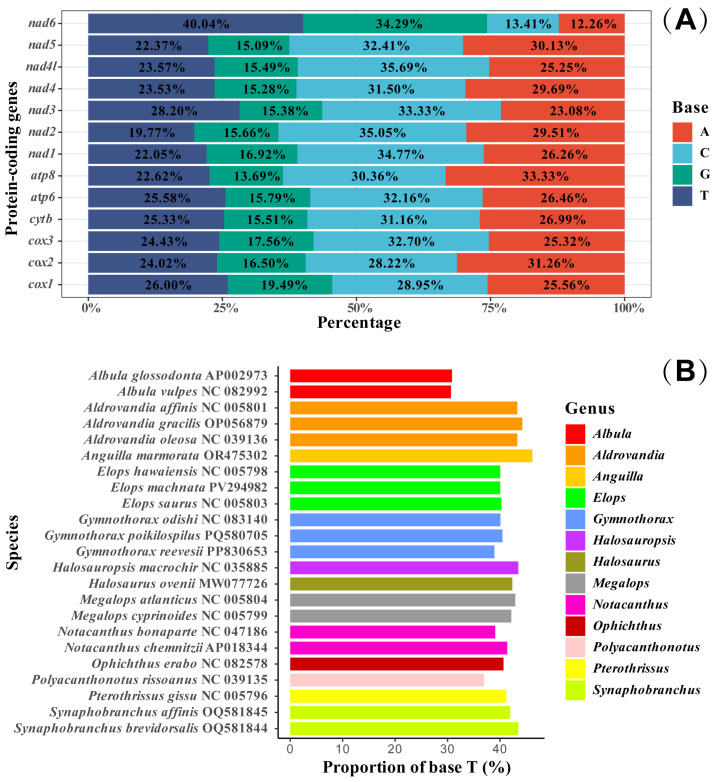
The proportion of various bases in 13 PCGs in *E. machnata* (**A**). The base composition of nad6 in the mitochondrial genomes of 23 Elopomorpha species (**B**).

**Figure 4 biology-14-00739-f004:**
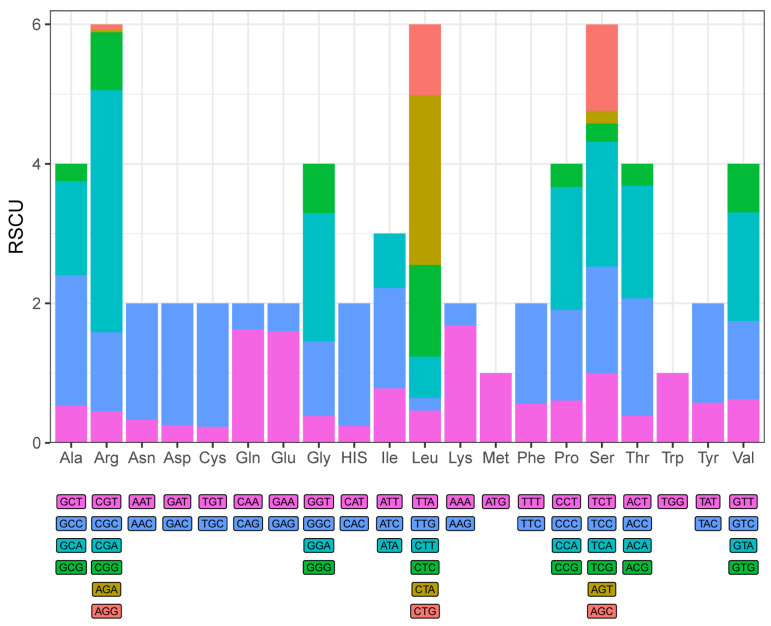
Relative synonymous codon usage values (RSCU) of 13 PCGs in *E. machnata*.

**Figure 5 biology-14-00739-f005:**
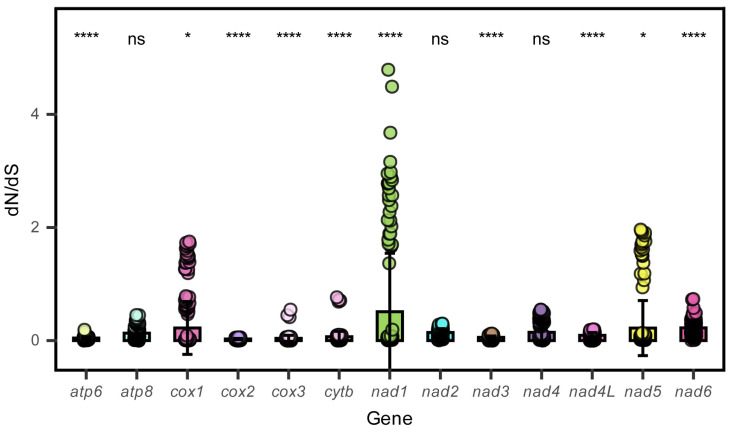
The ratio of non-synonymous (dN) to synonymous (dS) mutation rates for 13 PCGs in 23 Elopomorpha species. ns, not significant; *, *p* < 0.05; ****, *p* < 0.00001.

**Figure 6 biology-14-00739-f006:**
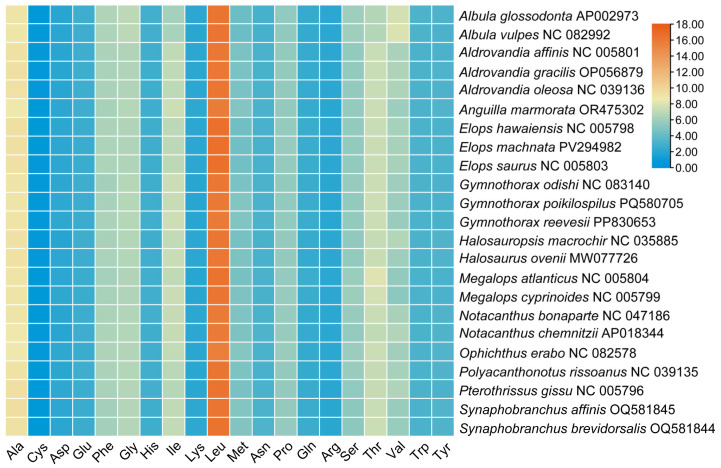
Amino acid composition of 13 protein-coding genes from 23 species of Elopomorpha.

**Figure 7 biology-14-00739-f007:**
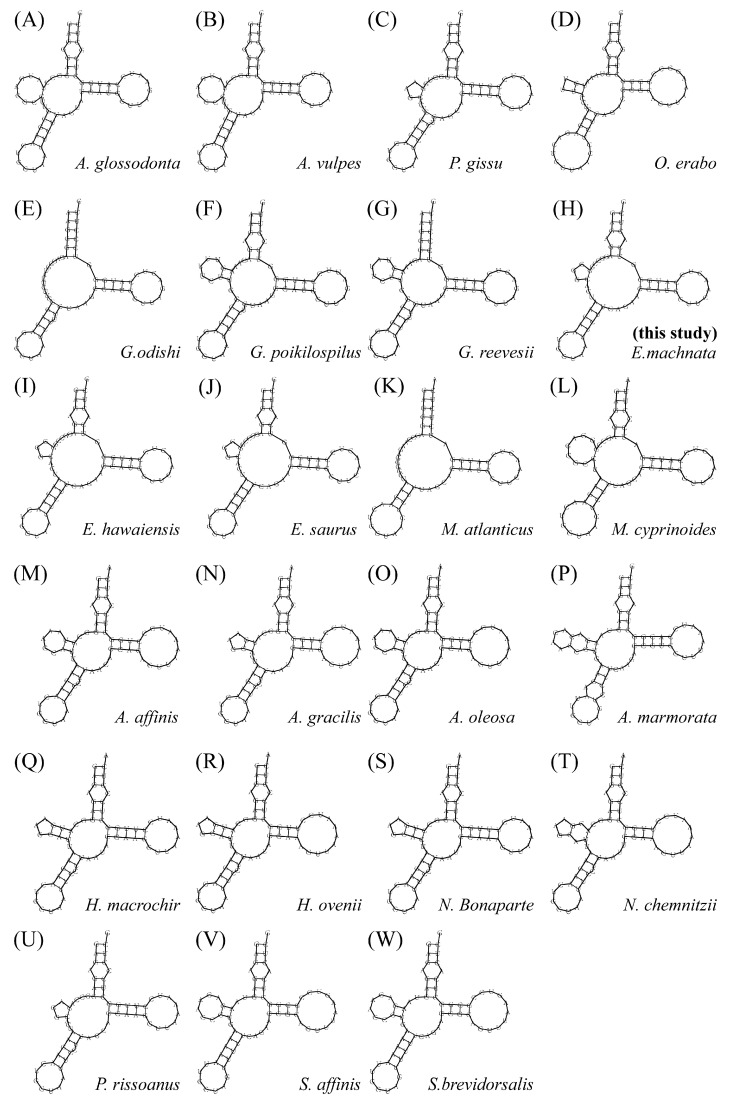
Secondary structure diagrams of the trnS1(gct) gene from 22 species within the superorder Elopomorpha, showing Albuliformes (**A**–**C**); Anguilliformes (**D**–**G**); Elopiformes (**H**–**L**); and Notacanthiformes (**M**–**W**).

**Figure 8 biology-14-00739-f008:**
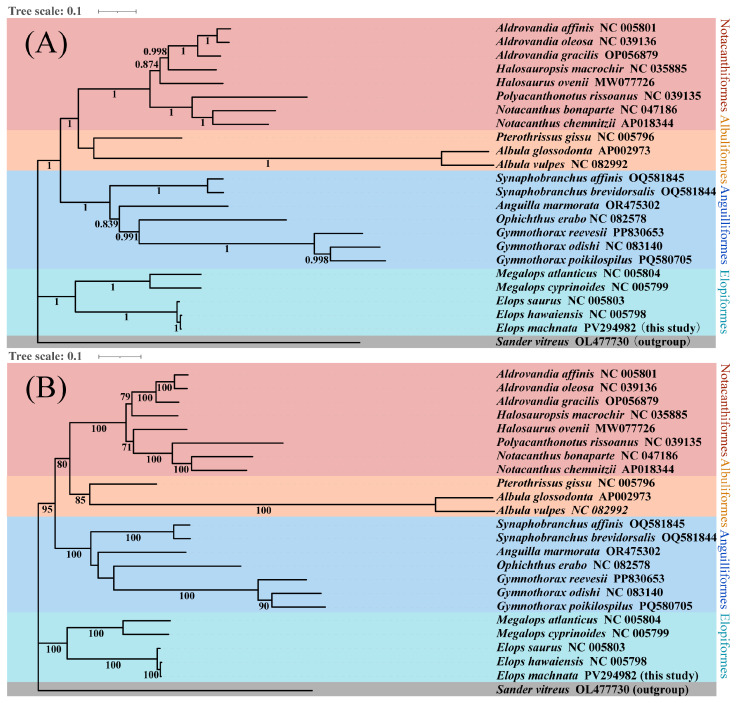
BI (**A**) and ML (**B**) phylogenetic trees reconstructed using 13 protein-coding sequences of Elopomorpha species. BI posterior probabilities and ML bootstrap support values are labeled near the nodes. *S. vitreus* was used as the outgroup.

**Table 1 biology-14-00739-t001:** The annotation results of the complete mitochondrial genome of *E. machnata*. Start = genomic coordinate where a gene begins (bp); Stop = genomic coordinate where a gene ends (bp); Start Coding = first nucleotide of the coding sequence (CDS); Stop Coding = last nucleotide of the coding sequence (CDS); Interval = non-coding region between adjacent genes (bp); Strand = DNA strand orientation (+ = heavy [H] strand; − = light [L] strand); Gene Length = total length of the gene (base pairs (bp)).

Gene	Start	Stop	Strand	Length	Interval	Start Coding	Stop Coding
*trnP*(tgg)	1	70	+	70	3		
*trnT*(tgt)	74	146	−	73	0		
*cytb*	147	1287	−	1141	5	ATG	ACT
*trnE*(ttc)	1293	1361	+	69	0		
*nad6*	1362	1883	+	522	−4	ATG	TAA
*nad5*	1880	3721	−	1842	0	ATG	TAA
*trnL1*(tag)	3722	3794	−	73	1		
*trnS1*(gct)	3796	3863	−	68	0		
*trnH*(gtg)	3864	3932	−	69	0		
*nad4*	3933	5313	−	1381	−7	ATG	TCT
*nad4l*	5307	5603	−	297	0	ATG	TAA
*trnR*(tcg)	5604	5673	−	70	−2		
*nad3*	5672	6022	−	351	0	ATG	TAG
*trnG*(tcc)	6023	6094	−	72	−1		
*cox3*	6094	6879	−	786	−1	ATG	TAA
*atp6*	6879	7562	−	684	−10	ATG	TAA
*atp8*	7553	7720	−	168	1	ATG	TAA
*trnK*(ttt)	7722	7795	−	74	0		
*cox2*	7796	8486	−	691	14	ATG	CCT
*trnD*(gtc)	8501	8572	−	72	8		
*trnS2*(tga)	8581	8651	+	71	−9		
*cox1*	8643	10,238	−	1596	1	GTG	AGG
*trnY*(gta)	10,240	10,310	+	71	0		
*trnC*(gca)	10,311	10,378	+	68	37		
*trnN*(gtt)	10,416	10,487	+	72	1		
*trnA*(tgc)	10,489	10,557	+	69	1		
*trnW*(tca)	10,559	10,630	−	72	−2		
*nad2*	10,629	11,675	−	1047	0	ATG	TAG
*trnM*(cat)	11,676	11,744	−	69	−1		
*trnQ*(ttg)	11,744	11,814	+	71	−1		
*trnI*(gat)	11,814	11,885	−	72	8		
*nad1*	11,894	12,868	−	975	0	ATG	TAA
*trnL2*(taa)	12,869	12,943	−	75	0		
*rrnL*	12,944	14,606	−	1663	39		
*trnV*(tac)	14,646	14,717	−	72	0		
*rrnS*	14,718	15,674	−	957	0		
*trnF*(gaa)	15,675	15,744	−	70	0		
*D-loop*	15,745	16,712		968			

## Data Availability

The data supporting this study are openly accessible in the NCBI repository under the GenBank accession number: PV294982.

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
