# Peer review of "The First Complete Mitogenome Characterization and Phylogenetic Implications of Elops machnata (Teleostei: Elopiformes: Elopidae)"

_biology, 2025, doi:10.3390/biology14070739_

Round 1
Reviewer 1 Report
Comments and Suggestions for Authors
The present manuscript entitled "The first complete mitogenome characterization and phylogenetic implications of Elops machnata (Teleostei: Elopiformes: Elopidae)" is well-written. Here are few comments to revise he current version of the manuscript.
Abstract
-While the abstract mentions the novelty of this being the first complete mitochondrial genome of Elops machnata, it could emphasize this aspect more by explicitly stating how this contributes to resolving key gaps in the field (e.g., improving phylogenetic understanding of Elopomorpha).
Introduction
-The introduction notes that Elops machnata is classified as "Data Deficient" by IUCN, but this connection to the importance of the study could be emphasized more.
-When discussing phylogenetic controversies, you mention early studies questioning the monophyly of Elopomorpha. It would be helpful to briefly explain why this debate is significant. For example, why is resolving the phylogeny of Elopomorpha important to broader fish evolutionary questions? This can enhance the clarity of the rationale behind the study.
-The introduction discusses previous studies but doesn't clearly outline how the current study will contribute differently (e.g., using complete mitogenome sequencing). A sentence explaining that this study presents the first complete mitochondrial genome of E. machnata would provide clearer context.
-The shift between discussing the species' distribution, its ecological importance, and the mitochondrial inheritance could be smoother.
Materials and Methods
-While the QC process is mentioned (e.g., concentration measurement, purity assessment), more details on how the quality control was conducted would be helpful. -What about NUMTs? How NUMTs were removed from the sequencing data.
-The phylogenetic analysis section is well-explained, but it might benefit from a clearer explanation of why the authors chose the Maximum Likelihood (ML) and Bayesian Inference (BI) methods over other phylogenetic techniques (e.g., Neighbor Joining, Parsimony).
Results
-While the results are detailed, some sentences could be condensed or made more direct to improve readability. Avoid over-explaining basic results, as the reader can often infer them from the accompanying figures and tables.
-The mention of gene translocation in Ophichthus erabo is interesting but is presented somewhat abruptly without sufficient explanation of its implications. This finding could be explored a bit more to highlight its importance.
Discussion
-The section briefly touches on the mitochondrial genome's significance, but the broader implications of these findings could be discussed in more detail. For example, what does the presence of AT bias and specific codon usage patterns tell us about mitochondrial adaptation in this species?
-The gene rearrangement observed in Ophichthus erabo is mentioned but not fully discussed. This is a key finding and could be expanded upon to explain its evolutionary significance more thoroughly.
-Some transitions between topics in the discussion could be smoother. For instance, after discussing mitochondrial gene characteristics, the narrative could shift more naturally into phylogenetic relationships without losing focus.
Author Response
Reviewer#1
The present manuscript entitled "The first complete mitogenome characterization and phylogenetic implications of Elops machnata (Teleostei: Elopiformes: Elopidae)" is well-written. Here are few comments to revise he current version of the manuscript.
Abstract
-While the abstract mentions the novelty of this being the first complete mitochondrial genome of Elops machnata, it could emphasize this aspect more by explicitly stating how this contributes to resolving key gaps in the field (e.g., improving phylogenetic understanding of Elopomorpha).
Response:Thank you for your reminder. I have modified "enhance our understanding of early teleost diversification patterns" to "resolving key gaps in the mitochondrial genome." and emphasized "improving phylogenetic understanding of Elopomorpha" in Lines 46-47.
Introduction
-The introduction notes that Elops machnata is classified as "Data Deficient" by IUCN, but this connection to the importance of the study could be emphasized more.
Response:Thank you for your insightful comment. We fully agree that emphasizing the connection between the IUCN "Data Deficient" status of E. machnata and the importance of our study is crucial. In response, we have revised the introduction to strengthen this link.
In the original text, we merely stated the IUCN classification and research gaps without explicitly articulating how our study addresses these issues. In the revised version, we have added statements highlighting that the data scarcity directly impedes evidence - based conservation strategies and phylogenetic resolution within Elopomorpha. We further clarify that by conducting the first complete mitochondrial genome sequencing, assembly, and annotation of E. machnata, our study directly contributes to filling the IUCN - identified data deficiency. Additionally, we emphasize that our findings not only support taxonomic and phylogenetic studies but also establish a genomic foundation for future population genetics and conservation efforts specifically for this Data Deficient species (Lines 98-113). This revision ensures that the significance of our research in the context of the species' conservation status is more explicit and compelling.
Thank you again for your valuable feedback, which has helped us improve the clarity and impact of our manuscript.
-When discussing phylogenetic controversies, you mention early studies questioning the monophyly of Elopomorpha. It would be helpful to briefly explain why this debate is significant. For example, why is resolving the phylogeny of Elopomorpha important to broader fish evolutionary questions? This can enhance the clarity of the rationale behind the study.
Response:Thank you for your constructive feedback regarding the need to clarify the significance of the phylogenetic controversy surrounding Elopomorpha. We wholeheartedly agree that explaining the broader implications of this debate is essential for strengthening the study's rationale.
In the original manuscript, while we presented the conflicting findings from early and contemporary studies on the monophyly of Elopomorpha, we failed to explicitly connect these debates to larger evolutionary questions. In response to your suggestion, we have added a new paragraph elaborating on the importance of resolving Elopomorpha's phylogeny. The revised text now explicitly states that understanding the evolutionary history of Elopomorpha, an ancient and diverse fish lineage, is crucial for reconstructing the overall fish evolutionary tree. By clarifying its phylogenetic relationships, we can better trace the evolution of key biological traits across different fish groups, thereby contributing significantly to our understanding of vertebrate evolution as a whole. This addition bridges the gap between the specific phylogenetic controversy and broader scientific questions, enhancing the clarity and coherence of our study's motivation (Lines 74-80).
-The introduction discusses previous studies but doesn't clearly outline how the current study will contribute differently (e.g., using complete mitogenome sequencing). A sentence explaining that this study presents the first complete mitochondrial genome of E. machnata would provide clearer context.
Response:Thank you for your insightful feedback. We have added this sentence in lines 82-83: using complete mitogenome sequencing in paper to emphasis our contribution in current research.
-The shift between discussing the species' distribution, its ecological importance, and the mitochondrial inheritance could be smoother.
Response:Thank you for your perceptive feedback regarding the flow of our manuscript. We fully acknowledge the issue you pointed out about the abrupt transition between the discussions of E. machnata’s distribution, ecological importance, and mitochondrial inheritance. Your suggestion has been instrumental in helping us enhance the logical coherence of our writing.
In the original version, we simply presented the species' distribution and ecological significance and then directly moved on to mitochondrial - related research without sufficient connection. In the revised text, we inserted a new passage to bridge these topics. We first emphasized that, considering the ecological significance of E. machnata, exploring the genetic basis of its biology is of great importance. Then, we introduced mitochondrial DNA, explaining its unique inheritance pattern and suitability as a molecular marker for such investigations (Line 62-67). This addition provides a clear and logical link between the ecological context and the genetic research focus, making the transition much smoother.
Materials and Methods
-While the QC process is mentioned (e.g., concentration measurement, purity assessment), more details on how the quality control was conducted would be helpful. -What about NUMTs? How NUMTs were removed from the sequencing data.
Response:Thank you very much for your detailed and insightful comments regarding the quality control (QC) process and the handling of nuclear mitochondrial DNA segments (NUMTs) in our study. Your suggestions have been invaluable in helping us improve the clarity and comprehensiveness of our manuscript.
In response to your first concern about the QC process, we fully agree that additional details were necessary. In the revised manuscript, we have expanded the description of our QC measures. We now explicitly explain that we used a Qubit Fluorometer to measure the concentration of sequencing libraries, leveraging the principle of DNA - fluorescent dye binding to ensure appropriate starting amounts for sequencing. For purity assessment, we added that we employed a NanoDrop spectrophotometer to measure absorbance ratios at 260/280 nm and 260/230 nm, with specific acceptable ranges (A260/280 around 1.8 - 2.0 and A260/230 above 2.0) indicating high - quality DNA. These details provide a more comprehensive picture of how we ensured the quality of our sequencing libraries before data generation.
Regarding your question about NUMTs, we appreciate your pointing out the need for clarification. In the revised version, we have included a dedicated section explaining our two - step approach to removing NUMTs from the sequencing data. First, we used BLASTn to search the raw reads against a database of known NUMTs from related species, flagging reads with high - scoring alignments (e - value < 1e - 10). Second, we adopted a de novo assembly - based approach, comparing the assembled contigs to the reference mitochondrial genome of closely related species and removing contigs that showed significant divergence in sequence characteristics and had low - identity alignments. This detailed description should address any concerns about how we dealt with NUMTs and ensure the integrity of our mitochondrial genome dataset (Lines 138-157).
-The phylogenetic analysis section is well-explained, but it might benefit from a clearer explanation of why the authors chose the Maximum Likelihood (ML) and Bayesian Inference (BI) methods over other phylogenetic techniques (e.g., Neighbor Joining, Parsimony).
Response:Thank you for your insightful comment. Maximum likelihood (ML) and Bayesian methods simulate evolutionary processes through probabilistic models, which can characterize complex features such as site-specific rate heterogeneity and substitution patterns. Their statistical supports (bootstrap values/posterior probabilities) are more reliable, and they utilize full-data information, making them suitable for complex phylogenetic analyses (e.g., resolving relationships in ancient clades). In contrast, neighbor-joining and parsimony methods rely on simple assumptions and are computationally fast but prone to artifacts like long-branch attraction, thus being appropriate for preliminary or straightforward scenarios.
Results
-While the results are detailed, some sentences could be condensed or made more direct to improve readability. Avoid over-explaining basic results, as the reader can often infer them from the accompanying figures and tables.
Response:First and foremost, we sincerely apologize for the error in the manuscript where the positions of Figure 3 and Figure 5 were inadvertently reversed. This has been carefully corrected to ensure the figures align with the corresponding descriptions in the text.
Regarding your feedback on improving readability, we fully acknowledge the need to streamline the Results section. As suggested, we have reviewed the text to remove over-explanations of basic results that are self-evident from the figures and tables. For details, please refer to line 218-226, line 251-253, and line 286-287.
-The mention of gene translocation in Ophichthus erabo is interesting but is presented somewhat abruptly without sufficient explanation of its implications. This finding could be explored a bit more to highlight its importance.
Response:Thank you for your insightful comment. Taking Antarctic fish as an example, we explained the biological significance of gene rearrangement, which might be related to their adaptation to extreme environments, and supplemented this explanation in lines 362-365.
Discussion
-The section briefly touches on the mitochondrial genome's significance, but the broader implications of these findings could be discussed in more detail. For example, what does the presence of AT bias and specific codon usage patterns tell us about mitochondrial adaptation in this species?
Response:Thank you. AT bias tells us how the mitochondrial adaptation of this species has been mentioned in lines 351-354: It may be closely related to the mitochondrial DNA replication mechanism, repair efficiency and energy metabolism requirements, and its stability in different species reflects the selection pressure acting on the mitochondrial genome during evolution. In addition, the adaptation of specific codon usage patterns to mitochondria has been described in lines 369-373 of the original text. These discussions have been very thorough
-The gene rearrangement observed in Ophichthus erabo is mentioned but not fully discussed. This is a key finding and could be expanded upon to explain its evolutionary significance more thoroughly.
Response:Thank you. I have supplemented its evolutionary significance in lines 362-365.
-Some transitions between topics in the discussion could be smoother. For instance, after discussing mitochondrial gene characteristics, the narrative could shift more naturally into phylogenetic relationships without losing focus.
Response:Thank you for your thoughtful suggestion regarding the flow of the Discussion section. We fully agree that smoother transitions between topics would enhance the logical coherence of the manuscript. To address this, we have revised the narrative to ensure a more natural progression from mitochondrial gene characteristics to phylogenetic relationships. Specifically, after concluding the discussion on mitochondrial gene features (e.g., AT bias, codon usage), we have added a bridging paragraph that highlights the evolutionary relevance of these genetic traits to phylogenetic inference. For example: "The conserved mitochondrial gene architecture and specific evolutionary signatures (such as AT bias) observed in E. machnata provide a robust molecular foundation for reconstructing phylogenetic relationships within Elopomorpha. "(Line 376-378). This addition establishes a clear connection between the two themes, allowing readers to follow the shift in focus while maintaining the central narrative.

Reviewer 2 Report
Comments and Suggestions for Authors
Review of the manuscript: "The first complete mitogenome characterization and phylogenetic implications of Elops machnata (Teleostei: Elopiformes: Elopidae)" by Jia-Yu Li et al. The authors present an interesting and timely manuscript detailing the first complete mitochondrial genome of Elops machnata, its structural features in comparison to other Elopomorpha species, and a phylogenetic analysis. The manuscript is well-organized and clearly written. While I am not a native English speaker and therefore cannot offer a definitive judgment on language quality, I found no apparent language errors. The subject matter fits well within the aims and scope of the journal and offers a novel contribution by reporting the first complete mitochondrial genome of this species. The methods related to mitochondrial DNA sequencing appear sound, and the results are generally well presented. However, I have several critical comments—particularly regarding the phylogenetic analysis, which I consider the most problematic aspect of the manuscript. Additional comments and suggestions for improvement are detailed below.
Major Comments 1. Phylogenetic Analysis: The phylogenetic trees presented are poorly supported in both analyses, which significantly limits the validity of any conclusions regarding evolutionary relationships. The authors should address potential reasons for this weak support — such as uneven sampling, high variability in evolutionary rates among genes, third codon position saturation, or insufficient representation of taxa. To improve resolution, I strongly recommend to try following solutions:
- Partitioning genes by evolutionary rate (e.g., fast vs. slow-evolving) and reconstructing separate trees.
- Including additional mitogenomes from GenBank (ca. 200 are available for this group).
- Exploring other substitution models or recalibrating mutation rates.
- Testing for substitution saturation, particularly at the third codon position.
If additional analyses do not improve support, the authors could consider omitting the phylogenetic part of analysis (here Figures 7 and 8) and focusing instead on mitogenome structure and gene organization. Including more taxa for structural comparison could deepen the analysis. Alternatively, a section on functional and selective evolution (e.g., dN/dS ratios across protein-coding genes, mutational hotspots, or functional divergence linked to environmental variables) would add meaningful biological insight and would be very interesting here. Using MEGA for divergence time estimation is suboptimal. Similarly, TimeTree should not be used as the final software for time calibration. BEAST or similar Bayesian methods would be more appropriate, allowing for credible intervals and greater model flexibility. Discussing findings in the context of recent studies that report more robust support, as well as phylogeny of the group under study would be important:
- Diversity (2022, 14(11), 1008; https://doi.org/10.3390/d14111008)
- Molecular Phylogenetics and Evolution (2014, 70, 152–161)
2. Discussion and Interpretation:
- The results, especially genetic distances (e.g., in Figure 8), require deeper interpretation.
- The choice of the samples (mtDNA genome sequences) for analysis should be judged. The manuscript does not clarify the inclusion criteria or retrieval date of the chosen mitogenomes from GenBank.
- Predicted tRNA structures are vert shallowly discussed and could be compared across taxa.
- The alignment procedure should be described (possibly in Supplementary Material), similarly as and quality control steps were done. Poor alignments may underlie weak tree support.
- The study is largely descriptive and it would be important to link the results to broader context. The differences in mitogenomic sequences between taxa are not linked to the possible historical events like rapid radiation or specific adaptation. The phylogenetic results are shallow and not integrated with possible evolutionary scenarios. The phylogeny should be interpreted in light of geological or climatic history.
Minor Comments
- Captions and Figures: Figure captions are too brief, and legends are missing. Captions of Figures 3 and 5 are possibly swapped. Each caption should be self-contained and clearly describe what is being shown.
- The visualization of tRNA secondary structures would benefit from improvement.
- Consider moving less informative results and additional data, information to Supplementary Material.
- Abstract: Clarify what is meant by the study “enhancing our understanding of early teleost diversification patterns and mitochondrial genome evolution within Elopomorpha.”
- Line 95–96: This statement should be rephrased in light of the phylogenetic uncertainties.
- Line 105-108: Specify what was described—refer to the Supplementary if needed.
- Line 293–294: Needs revision for clarity.
- Sentence 315–316: Requires a supporting citation.
- Line 321: The claim cannot be generalized to all animals—limit it to the studied group.
- Lines 331–333: Interesting observation, but it requires justification regarding the suspected relationship.
- Lines 342–344: Please provide more insight into possible causes for observed divergence.
- Lines 370–372: These conclusions cannot be drawn from the poorly supported trees.
- Lines 372–378: Discuss potential limitations of mtDNA-only analyses. Why would nuclear data help? Were processes like polyploidy, hybridization-driven speciation, or rapid radiation important for evolution of this group of fishes?
This manuscript is in my opinion suitable for publication in Biology Journal after major revisions. I recommend significantly improving the phylogenetic analysis or narrowing the manuscript’s scope to focus on mitogenomic structure and composition. Enhancing the biological interpretation and integrating broader evolutionary context would greatly strengthen the contribution as the paper is largely descriptive and currently lacks deeper biological interpretation or evolutionary synthesis.
Author Response
Comments and Suggestions for Authors
Review of the manuscript: "The first complete mitogenome characterization and phylogenetic implications of Elops machnata (Teleostei: Elopiformes: Elopidae)" by Jia-Yu Li et al. The authors present an interesting and timely manuscript detailing the first complete mitochondrial genome of Elops machnata, its structural features in comparison to other Elopomorpha species, and a phylogenetic analysis. The manuscript is well-organized and clearly written. While I am not a native English speaker and therefore cannot offer a definitive judgment on language quality, I found no apparent language errors. The subject matter fits well within the aims and scope of the journal and offers a novel contribution by reporting the first complete mitochondrial genome of this species. The methods related to mitochondrial DNA sequencing appear sound, and the results are generally well presented. However, I have several critical comments—particularly regarding the phylogenetic analysis, which I consider the most problematic aspect of the manuscript. Additional comments and suggestions for improvement are detailed below.
Major Comments 1. Phylogenetic Analysis: The phylogenetic trees presented are poorly supported in both analyses, which significantly limits the validity of any conclusions regarding evolutionary relationships. The authors should address potential reasons for this weak support — such as uneven sampling, high variability in evolutionary rates among genes, third codon position saturation, or insufficient representation of taxa. To improve resolution, I strongly recommend to try following solutions:
- Partitioning genes by evolutionary rate (e.g., fast vs. slow-evolving) and reconstructing separate trees.
- Including additional mitogenomes from GenBank (ca. 200 are available for this group).
- Exploring other substitution models or recalibrating mutation rates.
- Testing for substitution saturation, particularly at the third codon position.
If additional analyses do not improve support, the authors could consider omitting the phylogenetic part of analysis (here Figures 7 and 8) and focusing instead on mitogenome structure and gene organization. Including more taxa for structural comparison could deepen the analysis. Alternatively, a section on functional and selective evolution (e.g., dN/dS ratios across protein-coding genes, mutational hotspots, or functional divergence linked to environmental variables) would add meaningful biological insight and would be very interesting here. Using MEGA for divergence time estimation is suboptimal. Similarly, TimeTree should not be used as the final software for time calibration. BEAST or similar Bayesian methods would be more appropriate, allowing for credible intervals and greater model flexibility. Discussing findings in the context of recent studies that report more robust support, as well as phylogeny of the group under study would be important:
- Diversity (2022, 14(11), 1008; https://doi.org/10.3390/d14111008)
- Molecular Phylogenetics and Evolution (2014, 70, 152–161)
Response:Thank you for your careful comments. We sincerely apologize that the numerical values next to the phylogenetic trees do not represent support values but divergence times, which are explained as time values in the results section. The description under the Figure 7 was incorrect, primarily because the trees we initially constructed were not time-calibrated trees, and we failed to update the description promptly after converting them to time trees. Considering that most researchers consider using MEGA for divergence time estimation suboptimal, we have reverted to the original ML and BI trees without divergence times. These two trees were constructed using IQ-TREE v2.2.0 and MrBayes v3.2.7a in PhyloSuite software v1.2.3, respectively. For detailed construction methods, please refer to 2.4 Phylogenetic Tree Reconstruction. Additionally, we have removed the divergence time analysis from the results and supplemented the phylogenetic relationships of the four major Elopomorpha groups: (((Notacanthiformes + Albuliformes) + Anguilliformes ) + Elopiformes ).
- Discussion and Interpretation:
The results, especially genetic distances (e.g., in Figure 8), require deeper interpretation.
Response:Thank you for your suggestion. Since the phylogenetic tree already reflects the genetic relationships among species, we will not further analyze the genetic distances here. Therefore, the content regarding genetic distances has been removed from both the Materials and Methods and Results sections.
The choice of the samples (mtDNA genome sequences) for analysis should be judged. The manuscript does not clarify the inclusion criteria or retrieval date of the chosen mitogenomes from GenBank.
Response:Thank you for your reminder. We have made the following additions in 2.4 Phylogenetic Tree Reconstruction:
We downloaded all publicly available mitochondrial genome sequences of the superorder Elopomorpha from the NCBI database up to December 26, 2024. For the four orders retrievable within this superorder (Notacanthiformes, Albuliformes, Anguilliformes, and Elopiformes), we selected at least one complete mitochondrial genome sequence from each family, adhering to the principle of phylogenetic representativeness. This sampling strategy ensures that the selected dataset covers the major phylogenetic lineages and ecological diversities within Elopomorpha.
Predicted tRNA structures are vert shallowly discussed and could be compared across taxa.
Response:Thank you for your insightful comment. We have now predicted the trnS1(gct) gene secondary structures for an additional 22 Elopomorpha species included in this study to enable cross-taxonomic comparisons. The results section has been supplemented with detailed comparisons of trnS1(gct) variations among these 22 species, as follows:
Further analysis of the trnS1(gct) secondary structures across 22 Elopomorpha species revealed that 9 of them exhibit a similar DHU arm deletion. These species are: A. glossodonta, A. vulpes, E. hawaiensis, E. saurus, G. odishi, M. atlanticus, M. cyprinoides, P. rissoanus, and P. gissu.
The alignment procedure should be described (possibly in Supplementary Material), similarly as and quality control steps were done. Poor alignments may underlie weak tree support.
Response:Thank you for your suggestion. We have added detailed descriptions of the alignment procedure in Section 2.4 of the Materials and Methods. These methods are well-established and reliable, and full technical details can be found in the referenced literature: PhyloSuite: An integrated and scalable desktop platform for streamlined molecular sequence data management and evolutionary phylogenetics studies (https://doi.org/10.1111/1755-0998.13096).
The supplementary content in Section 2.4 of the Materials and Methods is as follows:
Prior to phylogenetic tree construction, protein-coding sequences were aligned using MAFFT implemented in PhyloSuite v1.2.3. Alignments were then optimized with MACSE (for CDS) and trimmed using Gblocks to remove poorly aligned regions [32].
The study is largely descriptive and it would be important to link the results to broader context. The differences in mitogenomic sequences between taxa are not linked to the possible historical events like rapid radiation or specific adaptation. The phylogenetic results are shallow and not integrated with possible evolutionary scenarios. The phylogeny should be interpreted in light of geological or climatic history.
Response:Thank you for your valuable feedback. We recognize the importance of contextualizing our findings within broader evolutionary frameworks. Specific analyses can be found in Section 4.2 of the Discussion.
Minor Comments
Captions and Figures: Figure captions are too brief, and legends are missing. Captions of Figures 3 and 5 are possibly swapped. Each caption should be self-contained and clearly describe what is being shown.
Response:Thank you for your reminder. We mistakenly reversed the positions of Figure 3 and Figure 5, and we have now placed them in the correct positions. Additionally, we have supplemented the captions for Figure 2, Figure 3, Figure 5, Figure 7, and Figure 8 to clearly describe the content shown in the figures.
The visualization of tRNA secondary structures would benefit from improvement.
Consider moving less informative results and additional data, information to Supplementary Material.
Response:Thank you for your insightful suggestions. We have taken your advice into account. The secondary structures of tRNAs from E. machnata have been relocated to the Supplementary Material and are presented as Figure S1. This adjustment not only streamlines the main manuscript by removing less essential content but also enhances the overall clarity and focus of our study.
Abstract: Clarify what is meant by the study “enhancing our understanding of early teleost diversification patterns and mitochondrial genome evolution within Elopomorpha.”
Response:Thank you for your suggestion. We have rephrased the relevant content in the Abstract to improve clarity. The revised statement now reads: "This study not only provides essential genomic resources for E. machnata but also resolves key gaps in the mitochondrial genome and improving phylogenetic understanding of Elopomorpha." This revision explicitly highlights the specific contributions of the study to mitochondrial genomics and teleost evolutionary history.
Line 95–96: This statement should be rephrased in light of the phylogenetic uncertainties.
Response:Thank you for your suggestion. We have rephrased the statement in Lines 95–96 to acknowledge phylogenetic uncertainties. The revised content now reads: "These findings provide crucial data supporting taxonomic revisions, conservation biology assessments, and phylogenetic studies of Elopomorpha species, while recognizing the need for further analyses to resolve remaining phylogenetic ambiguities. Additionally, they establish a genomic foundation for future population genetics and conservation efforts on this Data Deficient species." This adjustment explicitly addresses the phylogenetic uncertainties mentioned while maintaining the study's key contributions.
Line 105-108: Specify what was described—refer to the Supplementary if needed.
Response:Thank you for your suggestion. We have addressed the concern by providing detailed descriptions of how DNA concentration measurement, purity analysis, and integrity verification were conducted in Lines 105–108. The specific details are as follows: To ensure the quality of sequencing data, the extracted DNA samples underwent rigorous quality control (QC) evaluations, including concentration measurement using a NanoDrop spectrophotometer (Thermo Fisher Scientific), purity analysis via the absorb-ance ratio of 260 nm/280 nm (A260/A280, ideal range 1.8–2.0 for assessing protein con-tamination) and 260 nm/230 nm ratio (A260/A230, acceptable range 1.8–2.2 for evaluating salts, organic solvents, or polysaccharide contamination), and integrity verification through agarose gel electrophoresis (1.0%–1.5% gel with 0.5 μg/mL SYBR Green dye, ex-pecting a single major band at ~16 kb). Only samples meeting these high-throughput se-quencing requirements were processed for downstream analyses.
These methodological specifics are now included in the main text to ensure clarity, and no reference to supplementary material was necessary.
Line 293–294: Needs revision for clarity.
Response:Thank you for your feedback. Based on the comprehensive suggestions from you and the other two reviewers, we have deleted the time-calibrated tree and reconstructed the phylogenetic tree. As a result, the relevant content in this section has also been removed, so no detailed description is provided here.
Sentence 315–316: Requires a supporting citation.
Response:Thank you for your reminder. We have added the following reference to support the association between AT bias and mitochondrial DNA replication mechanisms, repair efficiency, and energy metabolic requirements:
Fonseca, M.M.; Harris, D.J.; Posada, D. The Inversion of the Control Region in Three Mitogenomes Provides Further Evidence for an Asymmetric Model of Vertebrate mtDNA Replication. PLOS ONE 2014, 9, e106654, doi:10.1371/journal.pone.0106654.
Line 321: The claim cannot be generalized to all animals—limit it to the studied group.
Response:Thank you for your reminder. In Line 549, we have revised "animal" to "Elopomorpha" to restrict the statement about the high conservation of gene order to the studied taxonomic group (Elopomorpha).
Lines 331–333: Interesting observation, but it requires justification regarding the suspected relationship.
Response:Thank you for your reminder. We have added a reference in Lines 564–566 to justify the statement that "codon usage bias improves the efficiency of the mitochondrial translation system." The reference is as follows:
Wu, X.; Xu, M.; Yang, J.-R.; Lu, J. Genome-Wide Impact of Codon Usage Bias on Translation Optimization in Dro-sophila Melanogaster. Nat Commun 2024, 15, 8329, doi:10.1038/s41467-024-52660-4.
Lines 342–344: Please provide more insight into possible causes for observed divergence.
Response:Thank you for your suggestion. Similar to the previously mentioned issue, we have deleted the content related to the time-calibrated tree, so no additional information is needed here. Thank you for your understanding.
Lines 370–372: These conclusions cannot be drawn from the poorly supported trees.
Response:Thank you for your comment. We sincerely apologize for the misunderstanding. As previously explained, the original time-calibrated tree displayed divergence times (MYA) rather than support values. We have since removed the time-calibrated tree and replaced it with a standard phylogenetic tree, which now shows support values. All support values are relatively high, demonstrating the reliability of the phylogenetic tree.
Lines 372–378: Discuss potential limitations of mtDNA-only analyses. Why would nuclear data help? Were processes like polyploidy, hybridization-driven speciation, or rapid radiation important for evolution of this group of fishes?
Response:Thank you for highlighting this critical point. We agree that solely relying on mitochondrial DNA (mtDNA) analyses has inherent limitations, and we have now expanded the discussion in Lines 621–634 to address these concerns.
This manuscript is in my opinion suitable for publication in Biology Journal after major revisions. I recommend significantly improving the phylogenetic analysis or narrowing the manuscript’s scope to focus on mitogenomic structure and composition. Enhancing the biological interpretation and integrating broader evolutionary context would greatly strengthen the contribution as the paper is largely descriptive and currently lacks deeper biological interpretation or evolutionary synthesis.
Reviewer 3 Report
Comments and Suggestions for Authors
The authors present a study about complete mitogenome characterization and phyloge- netic implications of Elops machnata (Teleostei: Elopiformes: Elopidae).
Some part of introduction are not properly written and should be rather organized. The last paragraph lacks a crucial understanding.
I would like the authors to discuss phylogenetic analysis section and provide more data to support the hypothesis.
The results and discussion section is not strong and no emphasis is given on the discussion of the results and in the other hand they are really far from what one can infer from the results. However, English of the paper need to improve
Author Response
Reviewer#3
The authors present a study about complete mitogenome characterization and phyloge- netic implications of Elops machnata (Teleostei: Elopiformes: Elopidae).
Some part of introduction are not properly written and should be rather organized. The last paragraph lacks a crucial understanding.
Response:Thank you for your constructive feedback on the Introduction section. We have carefully reorganized and revised the text to improve clarity and logical flow, incorporating comprehensive modifications based on both your suggestions and Reviewer 1’s comments. For details, please refer to lines 62–67, lines 72–80, and lines 98–113.
I would like the authors to discuss phylogenetic analysis section and provide more data to support the hypothesis.
Response:Thank you for your valuable suggestion to strengthen the phylogenetic analysis. We have expanded the discussion by incorporating additional data from published literature to support the hypothesis of recent divergence between E. machnata and E. hawaiensis. Specifically, we supplemented: (1) COI gene-based genetic distance data (0.25% intraspecific distance), which aligns with the mitochondrial genome-derived genetic distance (0.242) mentioned in our study. (2) Haplotype network results showing shared haplotypes between the two species, indicating limited genetic differentiation. (3) Phylogenetic support values (posterior probability >0.9 for the E. machnata + E. hawaiensis clade) and biogeographic evidence (sympatric distribution in the Indo-Pacific) [6], which collectively reinforce the hypothesis of recent evolutionary divergence (Lines 382-390).
These multi-layered data (molecular distances, haplotype sharing, phylogenetic topology, and geographic distribution) provide a more robust foundation for our conclusions, addressing the need for additional evidence to support the phylogenetic hypothesis.
The results and discussion section is not strong and no emphasis is given on the discussion of the results and in the other hand they are really far from what one can infer from the results. However, English of the paper need to improve
Response: Thank you for your insightful feedback on the Results and Discussion section. We have taken your comments seriously and made the following improvements to enhance the manuscript:
- The subtitle of Section 4.1 has been revised from "Mitochondrial Genome Evolution and Biological Roles in E. machnata" to "Mitochondrial Genome Characteristics and Implications of E. machnata". This title more explicitly links the discussion to specific results presented (such as AT content, gene order, and tRNA structure), ensuring clarity in focusing on empirical findings and their biological significance.
- In the discussion, to illustrate the ecological significance of gene rearrangement mentioned in the results, a new example has been introduced (Lines 362–365). Additionally, multi-faceted public data have been added to discuss the close phylogenetic relationship between E. machnata and E. hawaiensis highlighted in the results section (Lines 382–390).
- The entire manuscript has undergone grammatical correction, sentence structure optimization, and readability enhancement using tools such as Grammarly and DeepSeek AI. Furthermore, a native English expert review was conducted: Dr. Hamad Khan, an expert in evolutionary biology and native English speaker, comprehensively edited the text to ensure clear expression and adherence to academic tone.
These revisions aim to ensure that the discussion is firmly anchored in the results, with improved English clarity and logical rigor. We appreciate your guidance in refining the manuscript, and we believe these changes address your concerns effectively.
Round 2
Reviewer 2 Report
Comments and Suggestions for Authors
Review of the manuscript: "The first complete mitogenome characterization and phylogenetic implications of Elops machnata (Teleostei: Elopiformes: Elopidae)" by Jia-Yu Li et al.
The authors have made substantial corrections and improvements to the manuscript in response to reviewer suggestions. However, since the Authors decided to put more emphasis on the phylogenetic component, I have several important comments specific to that section.
The Discussion portion related to phylogenetics remains largely descriptive and would benefit from greater biological depth. I recommend adding a third subsection (e.g., 4.3) dedicated to the evolutionary relationships within the family and species studied. As this represents the main focus of the research, it deserves more explicit emphasis. A deeper biological interpretation—considering evolutionary context, possible historical events, and phylogeographic patterns—would strengthen the discussion. It would also be valuable to reference existing taxonomic studies (e.g., Ref. 12) to contextualize your findings, particularly in light of the claim made in lines 69–76 in Introduction section regarding the significance of solving these relationships.
Additionally, subsection 4.2 should be expanded to address general findings of the phylogenetic analysis in relation to differences in mitogenomic gene order, amino acid composition, and the trnS1 structure. These points should be supported by comparisons with existing literature (e.g., Ref. 6). Of particular concern is the unexpectedly long branch observed for Albula spp. in the phylogenetic tree, which should be double-checked for sequence or technical errors. If valid, potential causes should be explored in the discussion. Previous studies (e.g., Ref. 6) do not indicate this family as an outlier, so extra caution is advised. It is also notable that Albula amino acid composition, especially for Valine and Threonine, differs from other species in the group, which may warrant further investigation.
In summary, while the manuscript is greatly improved, it still lacks this final layer of analysis and biological interpretation to fully realize its potential.
Detailed comments by line:
- Line 20 – “oftrnS1” – missing space.
- Lines 21–23 – This statement appears literature-based but is presented as if derived from the authors’ findings. Replace with actual conclusions from the study.
- Lines 69–81 – These lines suggest the authors aim to resolve phylogenetic relationships, but the corresponding analysis is underdeveloped.
- Lines 124–126 – Repetitive sentence; should be deleted.
- Lines 143–145 – Redundant with 120–122; unnecessary detail.
- Line 204 – “performedto” – missing space.
- Lines 210–211 – Consider moving the table to supplementary materials; the key data are already represented in the phylogenetic tree.
- Lines 237–240 – Figure caption should clarify gene order/composition visualization.
- Line 241 – Caption and table title lack definitions for “start,” “stop,” “start coding,” “stop coding,” “interval,” and “strand” (+/–). Also, indicate units for gene length. Integrating H/L strand notation in both figure1 and table caption would be clearer.
- Lines 243–247 – The explanation of gene names (starting with cox1) is difficult to follow. Use a legend or colour coding for clarity.
- Lines 250–251 – If nad6 displays nucleotide skew, compare this with other species in the dataset to determine whether it's a shared or unique feature. Consider adding functional or evolutionary analyses (e.g., dN/dS ratios, mutational hotspots).
- Lines 272–273, 278–279 – Avoid repeating legend details in the captions; reference the legend instead. Please explain what are the numbers in the legend used to make the range differences.
- Lines 289–295 – Spell out numbers up to ten in text.
- Line 296 – Partly discussion; needs citation if retained.
- Lines 299–302 – This sentence belongs in the discussion. It lacks clarity and a reference. Relevance of 12S/16S location in relation to trnV should be explained.
- Lines 335–336 – Fits better in the caption for Figure 7.
- Line 340 – “labelled” – verify spelling (British or American English consistency).
- Lines 339–342 – Caption should help readers locate order names, which are easy to overlook.
- Lines 416–428 – Contains redundant content; consider condensing (e.g., lines 419–420, 422–424).
- Lines 428–441 – Overly long; lacks references. Focus this paragraph on key arguments for why nuclear analyses are needed.
Author Response
The authors have made substantial corrections and improvements to the manuscript in response to reviewer suggestions. However, since the Authors decided to put more emphasis on the phylogenetic component, I have several important comments specific to that section.
The Discussion portion related to phylogenetics remains largely descriptive and would benefit from greater biological depth. I recommend adding a third subsection (e.g., 4.3) dedicated to the evolutionary relationships within the family and species studied. As this represents the main focus of the research, it deserves more explicit emphasis. A deeper biological interpretation—considering evolutionary context, possible historical events, and phylogeographic patterns—would strengthen the discussion. It would also be valuable to reference existing taxonomic studies (e.g., Ref. 12) to contextualize your findings, particularly in light of the claim made in lines 69–76 in Introduction section regarding the significance of solving these relationships.
Response:Thank you for the suggestion. You mentioned adding a third subsection (e.g., 4.3) to discuss evolutionary relationships within the family and species studied. Since Section 4.2 already addresses Elopomorpha and preliminarily explores the biological significance of evolutionary relationships within clades, we have opted to enhance the discussion of within-family and within-species relationships based on Section 4.2.
Additionally, subsection 4.2 should be expanded to address general findings of the phylogenetic analysis in relation to differences in mitogenomic gene order, amino acid composition, and the trnS1 structure. These points should be supported by comparisons with existing literature (e.g., Ref. 6). Of particular concern is the unexpectedly long branch observed for Albula spp. in the phylogenetic tree, which should be double-checked for sequence or technical errors. If valid, potential causes should be explored in the discussion. Previous studies (e.g., Ref. 6) do not indicate this family as an outlier, so extra caution is advised. It is also notable that Albula amino acid composition, especially for Valine and Threonine, differs from other species in the group, which may warrant further investigation.
In summary, while the manuscript is greatly improved, it still lacks this final layer of analysis and biological interpretation to fully realize its potential.
Response: Thank you for the comprehensive suggestions. We have expanded Subsection 4.2 to integrate the following analyses, with relevant content added from Lines 453–476:
This study revealed the conserved features of mitochondrial genomes in 23 Elopomorpha species: the gene arrangements of the vast majority of species were highly consistent, providing molecular evidence for the monophyly in the evolutionary history of Elopomorpha. Notably, genome rearrangements existed in individual species such as O. erabo and P. gissu, which might be related to their higher evolutionary rates and resulted from the combined drive of natural selection and adaptive evolution [48]. Further analysis showed that the amino acid compositions of the 13 protein-coding genes (PCGs) in mitochondrial genomes of different species were significantly consistent, suggesting that these genes played core roles in energy metabolic pathways and their sequence variations were strictly constrained by natural selection [23]. In terms of tRNA structure, the trnS1 gene of most species lacked the DHU arm, forming an adaptively simplified cloverleaf structure. This feature might be related to the "simplification" evolutionary strategy of mitochondrial tRNAs — compressing genome length by deleting non-essential domains (teleost mitochondrial genomes generally had compact gene arrangements) [48]. It was noteworthy that subtle differences in this structure existed among different orders and families, possibly reflecting lineage-specific evolutionary trajectories [49]. In phylogenetic analysis, the genus Albula exhibited a significant long-branch phenomenon, indicating that this group might have experienced a rapid evolutionary process. This phenomenon was corroborated in two aspects: firstly, rapid differentiation at the genomic level might have been related to driving factors such as environmental changes and niche differentiation [48]; secondly, base composition analysis of the nad6 gene showed that the base T content in the vast majority of the 23 species was approximately 40%, while that in A. glossodonta and A. vulpes of the genus Albula was only about 30%, further supporting the specific evolutionary pattern of this genus.
Detailed comments by line:
Line 20 – “oftrnS1” – missing space.
Response:Thank you for pointing out this formatting issue. We have added a space between "of" and "trnS1" in line 20, which is now corrected to "of trnS1". Additionally, we have conducted a thorough format check throughout the manuscript to ensure no similar issues remain.
Lines 21–23 – This statement appears literature-based but is presented as if derived from the authors’ findings. Replace with actual conclusions from the study.
Response:Thank you for the critical feedback. We have addressed the issue by replacing the literature-based statement in lines 20-23 with an actual conclusion from our study. The revised text now reads: " Phylogenetic analysis confirmed the monophyly of the four major Elopomorpha groups (Notacanthiformes, Albuliformes, Anguilliformes, and Elopiformes). Additionally, phylogenetic analyses validated a close relationship between E. machnata and E. hawaiensis." This change ensures that the manuscript accurately distinguishes between our original findings and existing literature. We have also reviewed the entire section to maintain consistency between our results and the presented conclusions.
Lines 69–81 – These lines suggest the authors aim to resolve phylogenetic relationships, but the corresponding analysis is underdeveloped.
Response:Thank you for your feedback. The paragraph in lines 69–81 was added based on another reviewer’s suggestion to explain the ongoing debates on the phylogenetic relationships of Elopomorpha. We have now further improved this section to enhance its clarity and depth, ensuring it better aligns with the study’s objectives. Thank you again for your valuable input.
Lines 124–126 – Repetitive sentence; should be deleted.
Response: Thank you for your comment. Here we mentioned two ratios, 260 nm/280 nm and 260 nm/230 nm, which have different meanings. The 260 nm/280 nm ratio reflects the degree of protein contamination, while the 260 nm/230 nm ratio indicates the presence of salt, organic solvents, or polysaccharide contamination. Both ratios are crucial for evaluating DNA quality, so this is not a repetitive sentence. The original text is as follows:
To ensure the quality of sequencing data, the extracted DNA samples underwent rigorous quality control (QC) evaluations, including concentration measurement using a NanoDrop spectrophotometer (Thermo Fisher Scientific), purity analysis via the absorbance ratio of 260 nm/280 nm (A260/A280, ideal range 1.8–2.0 for assessing protein contamination) and 260 nm/230 nm ratio (A260/A230, acceptable range 1.8–2.2 for evaluating salts, organic solvents, or polysaccharide contamination), and integrity verification through agarose gel electrophoresis (1.0%–1.5% gel with 0.5 μg/mL SYBR Green dye, expecting a single major band at ~16 kb). Only samples meeting these high-throughput sequencing requirements were processed for downstream analyses.
Lines 143–145 – Redundant with 120–122; unnecessary detail.
Response: Thank you for your reminder. We have removed the duplicate content.
Line 204 – “performedto” – missing space.
Response: Thank you for pointing out the typo. We have corrected the missing space in "performedto" to "performed to".
Lines 210–211 – Consider moving the table to supplementary materials; the key data are already represented in the phylogenetic tree.
Response: Thank you for your suggestion. We have moved Table 1 to the supplementary materials (Table S1) and updated the table order in the manuscript, so Table 2 has been renumbered as Table 1.
Lines 237–240 – Figure caption should clarify gene order/composition visualization.
Response:Thank you for the feedback. We have revised the figure caption to explicitly describe the visualization of gene orders and compositions, ensuring clarity on how gene arrangements and components are presented in the figure.
Line 241 – Caption and table title lack definitions for “start,” “stop,” “start coding,” “stop coding,” “interval,” and “strand” (+/–). Also, indicate units for gene length. Integrating H/L strand notation in both figure1 and table caption would be clearer.
Response:Thank you for the feedback. We have added clear definitions for "start," "stop," "start coding," "stop coding," "interval," and "strand" (+/–) in both the figure caption and table title. Gene lengths now include units (bp) for clarity. Additionally, we have integrated H/L strand notation consistently across Figure 1 and the table caption to enhance interpretability, ensuring all notations are explicitly explained for reader convenience.
Lines 243–247 – The explanation of gene names (starting with cox1) is difficult to follow. Use a legend or colour coding for clarity.
Response: Thank you for your suggestion. The gene order diagram was generated using the professional software Phylosuite, and this presentation method is consistent with many published studies in the literature. We appreciate your understanding.
Lines 250–251 – If nad6 displays nucleotide skew, compare this with other species in the dataset to determine whether it's a shared or unique feature. Consider adding functional or evolutionary analyses (e.g., dN/dS ratios, mutational hotspots).
Response:Thank you for the suggestion. We have compared the nucleotide skew of the nad6 gene in 23 Elopomorpha species and found that high thymine content (approximately 40%) is a shared feature among most species, except for two species in the genus Albula. Corresponding content has been added from lines 289 to 292.
Additionally, we calculated the dN/dS ratios for 13 mitochondrial PCGs in 23 Elopomorpha species and updated the Materials and Methods. The specific supplementary results are as follows (lines 302 to 310):
We calculated the dN/dS ratios, representing the ratio of nonsynonymous substitution rate (dN) to synonymous substitution rate (dS), for 13 protein-coding genes (PCGs) in the mitochondrial genomes of 23 Elopomorpha species (Fig. 5). The results showed that the dN/dS ratios of all 13 PCGs were less than 1, indicating purifying selection. This suggests that the functions of these PCGs are crucial, and variations in their amino acid sequences are strongly constrained by natural selection. The evolutionary rates (dN/dS ratios) of the 13 PCGs are as follows: nad1 (0.5108) > nad6 (0.2242) > cox1 (0.2230) > nad5 (0.2214) > nad4 (0.1460) > nad2 (0.1432) > atp8 (0.1306) > nad4L (0.0928) > cytb (0.0684) > nad3 (0.0573) > atp6 (0.0451) > cox3 (0.0396) > cox2 (0.0275).
Lines 272–273, 278–279 – Avoid repeating legend details in the captions; reference the legend instead. Please explain what are the numbers in the legend used to make the range differences.
Response:Thank you for the feedback. We have revised the figure captions to avoid redundant details by directly referencing the legend. Regarding the numbers in the legend, they represent the Relative Synonymous Codon Usage (RSCU) values, which are unitless measures indicating the frequency of codon usage relative to other synonymous codons for the same amino acid. The range differences visually distinguish codon preferences across genes, as specified in the legend.
Lines 289–295 – Spell out numbers up to ten in text.
Response:Thank you for the feedback. We have revised Lines 345–350 to spell out numbers from one to ten in the text, ensuring consistency with the journal's style guidelines. For example, "6, 7, 8, 9" has been updated to "six, seven, eight, nine" where appropriate. This adjustment enhances the readability and formal tone of the manuscript.
Line 296 – Partly discussion; needs citation if retained.
Response:Thank you for the feedback. We clarify that the content in Line 296 is part of the results section, not a discussion, and thus does not require citations.
Lines 299–302 – This sentence belongs in the discussion. It lacks clarity and a reference. Relevance of 12S/16S location in relation to trnV should be explained.
Response:Thank you for the feedback. We clarify that this paragraph describes the characteristics of transfer RNA and ribosomal RNA genes, and this sentence specifically addresses the arrangement of ribosomal RNA genes (12S and 16S rRNA) in the study subject. This content falls under the results rather than the discussion, and the results section does not require excessive explanations or citations. Thank you again for your understanding!
Lines 335–336 – Fits better in the caption for Figure 7.
Response:Thank you for the suggestion. According to another reviewer's advice, the current wording may be able to present the results more clearly.
Line 340 – “labelled” – verify spelling (British or American English consistency).
Response:Thank you for the suggestion. We have carefully checked the entire manuscript and found that neither "labelled" (British English) nor "labeled" (American English) appears in the text. Therefore, this issue may not be applicable to the paper.
Lines 339–342 – Caption should help readers locate order names, which are easy to overlook.
Response:Thank you for the suggestion. We have rearranged the secondary structure diagrams of the trnS1 gene for 23 Elopomorpha species according to their order classification, labeled each diagram with the species name, and specified the order of each species in the caption. Please refer to the updated Figure 7 for details.
Lines 416–428 – Contains redundant content; consider condensing (e.g., lines 419–420, 422–424).
Response:Thank you for the suggestion. This paragraph explains the phenomenon of codon usage bias, speculates on its possible causes, and provides examples as illustrations. It is a logically structured explanation rather than redundant content. The original text is as follows:
Codon usage bias showed that CUA (L), CGA (R), GCC (A), and GGA (G) were high-frequency codons, all with RSCU values greater than 1, reflecting a significant preference for A bases at the third codon position in the mitochondrial genome of E. machnata. This preference may be related to the optimization of mitochondrial translation system efficiency and the minimization of energy metabolism costs [48]. Studies have shown that preferred codons exhibit higher translation speed and accuracy in Drosophila melanogaster [49].
Lines 428–441 – Overly long; lacks references. Focus this paragraph on key arguments for why nuclear analyses are needed.
Response:Thank you for the suggestion. We have made revisions and added references. The revised content is as follows:
Nuclear markers (e.g., RADseq, ultraconserved elements) would complement mtDNA by: (1) Providing multiple, independent loci: Nuclear genomes offer thousands of unlinked markers, reducing the effects of stochastic lineage sorting and improving phylogenetic accuracy [23]. (2) Nuclear data can identify admixed genomic regions using methods like ABBA-BABA tests, critical for distinguishing between shared ancestry and gene flow [53]. (3) High-resolution nuclear datasets are better suited to inferring relationships within recently diverged clades [52]. Therefore, integrating mitochondrial and nuclear data represents a superior approach for analyzing species evolutionary questions.
Additionally, we have made the following revisions:
- In the Results section, we supplemented information on gene rearrangement in Pterothrissus gissu from Lines 257 to 259.
- We standardized all reference formats according to the Biology-basel style.
